# TokenDrop + BucketSampler: Towards Efficient Padding-free Fine-tuning of Language Models

**Amrit Nagarajan**
School of ECE, Purdue University
nagaraj9@purdue.edu

**Anand Raghunathan**
School of ECE, Purdue University
raghunathan@purdue.edu

## Abstract

The great success of Language Models (LMs) for various Natural Language Processing (NLP) tasks is accompanied by computational challenges during both pre-training and fine-tuning. Pre-training has attracted significant attention due to its huge computational footprint. We focus on the fine-tuning of pre-trained LMs, which is expected to be performed much more frequently as the pre-trained models are adapted to downstream tasks. During fine-tuning, the presence of variable-length input sequences necessitates the use of padding tokens when batching sequences. These padding tokens lead to ineffectual computations, adversely impacting the efficiency of fine-tuning. We also observe that LMs memorize the limited task-specific training data despite the use of known regularization methods. Based on these insights, we present TokenDrop + BucketSampler, a framework that simultaneously improves efficiency and accuracy of LM fine-tuning. BucketSampler generates batches of samples with lower variance in sequence lengths to reduce the number of padding tokens, but does so without the accompanying accuracy drop seen in previous approaches. TokenDrop is a new regularizer that prunes a random subset of insignificant tokens from each input sequence in every epoch to prevent overfitting. TokenDrop drops more tokens from the longer sequences in each batch to further reduce variance in input lengths and the need for padding. TokenDrop + BucketSampler accelerates fine-tuning on diverse downstream tasks by up to $10.61\times$, while also producing models that are up to 1.17% more accurate compared to conventional fine-tuning. Code is available at https://github.com/amrnag/TokenDrop-BucketSampler.

.

## 1 Introduction

Language Models (LMs) derived from attention-based Transformer networks have significantly advanced the field of Natural Language Processing (NLP), with recent models showing remarkable performance on a wide range of tasks (Bubeck et al., 2023). LMs are typically characterized by large model sizes, and are pre-trained on very large text corpora. Pre-trained LMs are subsequently fine-tuned to solve a range of downstream tasks. While pre-training is the most expensive step by far in LM creation, and hence has attracted the most attention, we observe that the relatively high frequency of fine-tuning makes it an important challenge in its own right. Data available from public-domain pre-trained LMs such as BERT (Devlin et al., 2019) and OPT (Zhang et al., 2022) suggests that these LMs have been fine-tuned millions of times by different users on a diverse range of downstream tasks. Further, LMs are notoriously sensitive to the initialization of the task-specific final layer and the order in which training data is presented during fine-tuning (Dodge et al., 2020). As a result, multiple runs of fine-tuning with different hyperparameters are often required to achieve high accuracy on a given downstream task. Finally, due to large model sizes of LMs (that continue to grow), even a single fine-tuning run is compute-intensive and can take several GPU-days. In summary, while the enormous computational costs of pre-training have attracted a lot of attention, it is likely that comparable if not more compute time and energy have been spent on fine-tuning compared to pre-training since the inception of LMs.

While several prior efforts have focused on reducing the costs of pre-training, relatively little attention has been paid to the computational challenges of fine-tuning. We observe that LM fine-tuning presents a unique set of challenges. First, fine-tuning is performed on variable-length text sequences, with significant spread in lengths. When batches of variable-length sequences are generated for fine-tuning, shorter sequences in a batch are "padded" to the length of the longest sequence in the batch by adding padding tokens. However,

computations on padding tokens are useless, and adversely affect throughput during fine-tuning. Second, fine-tuning is performed in a supervised manner and requires expensive human annotations. As a result, fine-tuning datasets are several orders of magnitude smaller than pre-training datasets. In addition, when overparameterized LMs are trained on small task-specific datasets, overfitting leads to sub-optimal generalization performance (Fig. 1), even with the use of popular regularizers such as Dropout (Srivastava et al., 2014). Consequently, fine-tuning is performed over a very small number of epochs (typically ≤5). Finally, we observe that fine-tuned LMs are adversely impacted by even minor grammatical errors in inputs during inference when only grammatically perfect sequences are used for fine-tuning (Fig. 2). This sensitivity is problematic since the assumption of seeing only grammatically correct sequences during inference does not always hold in real-world scenarios, especially when LM inputs are provided by users with different levels of language proficiency.

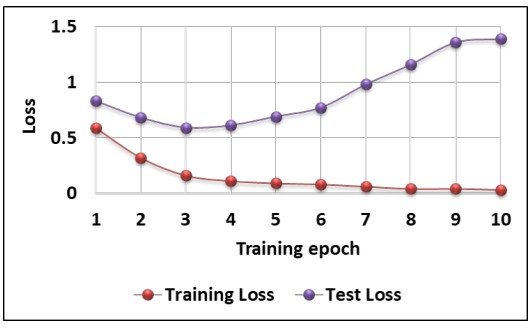

Figure 1: **Training curve obtained from fine-tuning Roberta-base on RTE, a language understanding task with 2.5K training samples, with dropout rate = 0.1.** We report loss averaged across 10 random seeds.

To address the aforementioned challenges, we present TokenDrop + BucketSampler, the synergistic combination of a new regularizer and batching method for accurate and efficient LM fine-tuning.

Our first contribution, BucketSampler, generates batches of samples with reduced spread in sequence lengths, thereby reducing computational overheads due to padding. In addition, the batch size is varied based on the lengths of the samples in the batch to maximize hardware utilization. Improved batching strategies have been incorporated into popular NLP libraries such as HuggingFace Transformers (Wolf et al., 2019) and Lingvo (Shen et al., 2019). However, these methods are optimized for training LMs from scratch. When applied to fine-tuning

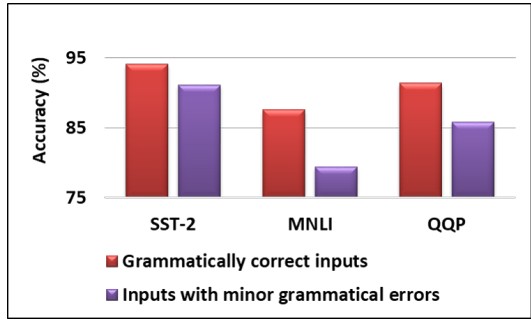

Figure 2: **Impact of minor grammatical errors on a Roberta-Base model fine-tuned using only grammatically correct examples.** We generate inputs with minor grammatical errors by pruning articles ('a', 'an', 'the') and punctuation marks (comma, fullstop, apostrophe, etc.) from samples in the development set.

on small datasets for very few epochs, we find that these prior batching methods lead to significant drop in accuracy. BucketSampler includes key algorithmic enhancements that tune the optimal batch size and learning rate schedules, thereby enabling fine-tuning to achieve high accuracy while also maintaining high hardware utilization.

Our second contribution is TokenDrop, a novel regularizer that identifies and drops a random subset of insignificant tokens from each sequence in every epoch. More tokens are dropped from the longer sequences in each batch, further reducing the need for padding (Fig. 3). Further, TokenDrop reduces overfitting by ensuring that the model does not see the same sequence repeatedly over the course of fine-tuning. As a side effect, it also improves the resilience of LMs to grammatical errors during inference by exposing the model to grammatically incorrect inputs during fine-tuning.

Since TokenDrop + BucketSampler improves fine-tuning efficiency by eliminating ineffectual computations, it can be combined with previously proposed approaches for parameter-efficient fine-tuning, such as freezing layers (Lee et al., 2019; Zaken et al., 2022) and the use of adapters (Houlsby et al., 2019), to achieve further efficiency gains. We summarize our main contributions as follows:

- We propose TokenDrop + BucketSampler, a framework for accurate and efficient LM fine-tuning.

- BucketSampler is a length-aware batching method that generates batches of sequences with similar lengths to reduce padding. BucketSampler incorporates optimizations that enable fine-tuning convergence while maintaining high

throughput.

- TokenDrop is a regularizer that drops a random subset of insignificant tokens from each sequence in every epoch to prevent LMs from memorizing fine-tuning data, while also further minimizing the need for padding.

- We demonstrate that TokenDrop can be synergistically combined with BucketSampler to simultaneously improve both accuracy (up to 1.2%) and efficiency (up to $10.61\times$) of LM fine-tuning.

## 2 Method

TokenDrop + BucketSampler is a framework for accurate and efficient fine-tuning of pre-trained LMs. The first component, TokenDrop, is a regularizer that randomly drops a subset of insignificant tokens from each sequence in every epoch. The second component, BucketSampler, is a length-aware batching method that generates batches of samples with lower spread in sequence lengths. We demonstrate how TokenDrop and BucketSampler can be synergistically combined (hence the name TokenDrop + BucketSampler) to simultaneously reduce overfitting *and* eliminate the need for padding tokens, resulting in both accuracy and speed improvements (Fig. 3).

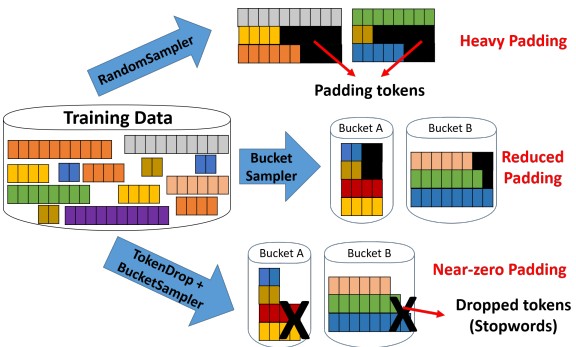

Figure 3: **Illustration of different batching methods.**

### 2.1 TokenDrop

Fine-tuning datasets for LMs are often small due to the need for expensive human annotations. As a result, we find that pre-trained LMs quickly memorize the training samples during fine-tuning (Fig. 1). Dropout (Srivastava et al., 2014), which drops a random subset of neurons in each training epoch, is currently the most widely used regularizer. However, we find that dropout is ineffective at preventing overfitting during fine-tuning of LMs (Fig. 1).

In addition, most fine-tuning datasets are only composed of sentences and phrases that are grammatically correct. As a result, minor errors in user-provided inputs during inference (such as missing punctuation marks) can degrade the quality of outputs produced by the LM (Fig. 2).

To address these challenges, we propose TokenDrop, a regularizer that drops a random subset of words from each sequence in every training epoch. Unlike Dropout, TokenDrop introduces *data diversity* between the different training epochs, thereby making the models unlikely to see the same sequence twice over the course of fine-tuning, and hence, less likely to overfit. We find that the choice of words selected for dropping has a significant impact on training convergence, since the semantics of the input text sequence can change if important tokens are dropped (see Appendix C). For instance, if the word "not" in the sequence "the movie was not good" is dropped, the sentiment of the sentence changes from negative to positive. In order to overcome this challenge, TokenDrop only drops stopwords from sequences. Stopwords are words in any language that do not contribute to the meaning of a sentence, but are added to make sentences grammatically correct. For instance, the words "the" and "was" in the aforementioned sequence are stopwords. In effect, TokenDrop provides a stronger regularization effect compared to Dropout (see Appendix D), and also improves the efficiency of fine-tuning by reducing the sequence length without affecting the meaning of the sequence. TokenDrop also improves the resilience of LMs to grammatical errors during inference by exposing the model to grammatically incorrect sequences during fine-tuning. In particular, TokenDrop generates grammatically incorrect sequences by pruning stopwords from grammatically correct sequences.

The procedure for applying TokenDrop to a given dataset is described in Algorithm 1. Given a dataset and a list of stopwords, we identify and prune a random subset of stopwords in each sequence. The number of stopwords pruned in each sequence is determined by the $Tokens\_to\_drop(sequence)$ parameter. When a global $TokenDrop\_Rate$ (analogous to $Dropout\_Rate$ in Dropout) is used, i.e., the same fraction of stopwords are dropped in each sequence, then $Tokens\_to\_drop(sequence) = number\_of\_stopwords(sequence) * TokenDrop\_Rate$. We note that it is not

necessary to use the same $TokenDrop\_Rate$ for all sequences, and a different fraction of stopwords can be dropped in each sequence, as described in Section 2.3. The complete list of stopwords used in our experiments is provided in Appendix C.

---

**Algorithm 1:** TokenDrop

---

**Input** : Input Dataset ($Dataset$), List of stopwords ($Stopwords$), Number of stopwords to drop in each sample ($Tokens\_to\_drop$)

**Output** : Dataset after applying TokenDrop ($Filtered\_Dataset$)

1   $Filtered\_Dataset = Dataset$
2   **for** $sample$ $in$ $Filtered\_Dataset$ **do**
3     $Stopwords$ = random_shuffle($Stopwords$)
4     **while** $Tokens\_to\_drop(sample) > 0$ **do**
5       **for** $word$ $in$ $Stopwords$ **do**
6         **if** $word$ $in$ $sample$ **then**
7           $sample$.pop($word$)
8           $Tokens\_to\_drop(sample)$ -= 1

9   return $Filtered\_Dataset$

---

## 2.2 BucketSampler

BucketSampler combines a sequence-length-aware, variable-batch-size batching strategy with algorithmic optimizations to enable faster fine-tuning. The method for generating batches with BucketSampler is illustrated in Algorithm 2. All samples in the dataset are divided into buckets such that sequences that fall into the same bucket have similar sequence lengths. Each bucket is defined by a triplet ($min\_seq\_len$, $max\_seq\_len$, $HWCap$). Here, $min\_seq\_len$ and $max\_seq\_len$ denote the minimum and maximum sequence lengths of sequences in the bucket, respectively, i.e., all sequences whose lengths lie between $min\_seq\_len$ (inclusive) and $max\_seq\_len$ (exclusive) of a bucket fall into that bucket (lines 20-23 in Alg. 2). Then, batches are generated by only combining sequences from the same bucket (lines 24-32 in Alg. 2). In effect, BucketSampler reduces the spread of sequence lengths in a batch, thereby reducing the need for padding tokens. BucketSampler also provides support for variable batch sizes. In particular, since the peak memory requirements for processing a batch scale quadratically with sequence length due to the quadratic complexity of self-attention, we propose using large batch sizes when generating batches from buckets with small $max\_seq\_len$ and viceversa. The $HWCap$ parameter associated with each bucket encodes the maximum batch size that can be used for generating batches from that bucket on a given hardware platform. $HWCap$ is experi-

mentally determined by profiling on-chip memory usage for different sequence lengths.

While the BucketSampler algorithm described above provides some improvements in the efficiency of fine-tuning, it still leaves room for further improvement along two directions. (1) The constraint that batches can only be formed using samples from the same bucket leads to "residual" batches with small numbers of samples, adversely affecting hardware utilization. Therefore, we propose Residual Batch Merging (RBM) to merge residual batches in different buckets into larger batches. (2) From an accuracy standpoint, we find that the convergence of fine-tuning is adversely impacted when using (a) very large batch sizes, and (b) a single learning rate schedule with variable batch sizes, since fine-tuning is often performed on small datasets for a very small number of epochs. To overcome these challenges, we propose Batch-Cap to progressively increase the maximum batch size over the course of fine-tuning. We also propose Learning Rate Modulation (LRM) to dynamically scale the learning rate based on the batch size. We explain these optimizations in greater detail in the following subsections.

**Residual Batch Merging (RBM):** Peak hardware utilization is achieved when the number of samples in each bucket is an exact multiple of the bucket's $HWCap$. However, in reality, it is highly likely that the aforementioned condition is not satisfied, resulting in one batch in most buckets with $batch\_size < HWCap$. We term these batches as "residual" batches, and the hardware is underutilized when processing residual batches. In order to reduce the impact of residual batches, we propose merging residual batches from different buckets into larger batches. We present the Residual Batch Merging (RBM) algorithm to maximize hardware utilization, while also minimizing the number of additional padding tokens introduced as a result of merging sequences from different buckets (lines 5-18 in Algorithm 2). New batches are created by appending samples one-by-one from residual batches in each bucket, with buckets processed in increasing order of $max\_seq\_len$ (lines 7, 11-12). When the number of samples in the new batch becomes an exact multiple of the $bucket\_batch\_size$ of the bucket corresponding to the longest sequence in this new batch, the new batch is added to the list of batches used for fine-tuning (lines 13-15). Lines 8-10 account for corner cases where the num-

---

**Algorithm 2:** BucketSampler

---

**Input** : Input Dataset ($Dataset$), Training epoch ($epoch$)
**Output** : Batches generated from $Dataset$ ($batches$)

1 **Function** BatchCap(*HWCap, epoch*):
2      EpochCap = $scaling\_factor^{epoch} \times$ base_batch_size
3      bucket_batch_size = min(HWCap, EpochCap)
4      return bucket_batch_size

5 **Function** Residual_Batch_Merging(*batches, residual_batches*):
6      new_batch = []
7      **for** *(residual_batch, bucket_batch_size) in residual_batches* **do**
8          **if** *cardinality(new_batch) >= bucket_batch_size* **then**
9              $batches$.append(new_batch)
10              new_batch = []
11          **for** *sequence in residual_batch* **do**
12              new_batch.append(sequence)
13              **if** *(cardinality(new_batch) > 0) and (cardinality(new_batch) % bucket_batch_size = 0)* **then**
14                  $batches$.append(new_batch)
15                  new_batch = []
16      **if** *new_batch is not empty* **then**
17          $batches$.append(new_batch)
18      return $batches$

19 $buckets$: List of buckets, where bucket is defined by the triplet (min_seq_len, max_seq_len, HWCap)
20 **for** *sample in $Dataset$* **do**
21      **for** *bucket in buckets* **do**
22          **if** *bucket.min_seq_len <= seq_len(sample) < bucket.max_seq_len* **then**
23              $bucket$.append($sample$)

24 $batches$ = []
25 residual_batches = []
26 **for** *bucket in buckets* **do**
27      $bucket$ = random_shuffle($bucket$)
28      bucket_batch_size = BatchCap($bucket$.HWCap, epoch)
29      index = 0
30      **while** *index < (floor_division(cardinality(bucket), bucket_batch_size) * bucket_batch_size)* **do**
31          $batches$.append($bucket$[index : index + bucket_batch_size])
32          index = index + bucket_batch_size
33      residual_batches.append(($bucket$[index :], bucket_batch_size))

34 $batches$ = Residual_Batch_Merging($batches$, residual_batches)
35 return $batches$

---

ber of samples in the new batch is less than the *bucket_batch_size* corresponding to the longest sequence in the new batch, but is larger than the *bucket_batch_size* corresponding to the next sequence to be added to the new batch. In this case, the new batch is considered to be "full", and is added to the list of batches used for fine-tuning. In effect, RBM reduces the number of possible residual buckets from $n$, where $n$ is the number of buckets, to 1 (only the final batch can be a residual batch, see lines 16-17). Since buckets are inspected in increasing order of $max\_seq\_len$, RBM also ensures that only residual samples from adjacent buckets are combined to form larger batches, thereby minimizing the number of padding tokens used in merged batches.

**BatchCap:** Since fine-tuning is performed on small datasets for small numbers of epochs, we find that fine-tuning does not converge when very large batch sizes are used due to the sparsity of weight updates. For instance, RTE (one of the GLUE (Wang et al.) datasets) contains 2.5K training samples, and fine-tuning on RTE is typically performed over 3 epochs. When a batch size of 1024 is used, only 3 weight updates are performed in each epoch, for a total of 9 weight updates during fine-tuning. As a result, fine-tuning does not converge, and models exhibit high training loss and low test accuracy at the end of fine-tuning. With BucketSampler, the $HWCap$ for buckets with small $max\_seq\_len$ can be large (for instance, $HWCap$ = 2525 for $max\_seq\_len$ = 5 when fine-tuning Roberta-Base on a NVIDIA RTX 2080Ti GPU). To ensure fine-tuning convergence,

BucketSampler uses smaller batch sizes in the first epoch of fine-tuning, and progressively increases the cap on the maximum allowable batch size in the later epochs of fine-tuning (Smith et al., 2018). We introduce BatchCap to find the maximum allowable batch size in each epoch (lines 1-4 in Alg. 2). In every epoch, BatchCap limits the maximum batch size of each bucket as $bucket\_batch\_size(bucket, epoch) = \min(HWCap(bucket), EpochCap(epoch))$, where $EpochCap(epoch) = scaling\_factor^{epoch} * base\_batch\_size$. Here, $base\_batch\_size$ and $scaling\_factor$ are hyperparameters that control the batch size used in the first epoch of fine-tuning, and the growth rate of the maximum batch size across epochs, respectively. We note that BatchCap leads to hardware under-utilization in the early epochs of fine-tuning. However, BatchCap is necessary to achieve convergence when fine-tuning with BucketSampler, and our exponential scaling rule ensures high utilization in the majority of epochs.

**Learning Rate Modulation (LRM):** The use of BucketSampler leads to large variance in batch sizes during fine-tuning. For instance, when fine-tuning Roberta-Base on a NVIDIA RTX 2080Ti GPU, $HWCap$ = 2525 for $max\_seq\_len$ = 5 and $HWCap$ = 64 for $max\_seq\_len$ = 128. Since the choice of learning rate is highly sensitive to the batch size used during training (Krizhevsky, 2014; Smith et al., 2018), we find that fine-tuning using BucketSampler fails to converge with a single learning rate schedule, even when a grid search is performed to find the best learning rate. We propose Learning Rate Modulation (LRM) to overcome the limitations of using a single learning rate schedule when training with variable batch sizes. LRM dynamically modulates the base learning rate based on the batch size of each batch. LRM scales the $base\_learning\_rate$ for each batch as $learning\_rate(batch) = base\_learning\_rate * sqrt(batch\_size(batch)/base\_batch\_size)$. Here, $base\_learning\_rate$ is the optimal learning rate schedule when training with fixed batch size, where all batches (except the last batch) have $batch\_size = base\_batch\_size$. The formula for computing $learning\_rate(batch)$ is derived from the square-root scaling law relating learning rate and batch size (Krizhevsky, 2014), which is a popular trick for parallelizing training of deep neural networks using large batch sizes on

GPU clusters. In particular, when batch size is changed to $k * batch\_size$, the learning rate is changed to $learning\_rate * sqrt(k)$ to find the optimal learning rate for a given batch size when the optimal learning rate is known for a different batch size. In this work, we use this formulation for modulating learning rate on a batch-by-batch basis, i.e., changing learning rate for each batch based on the size of the batch.

## 2.3 TokenDrop + BucketSampler

When batches are generated using BucketSampler, the sequence lengths of all samples in a batch lie between the $min\_seq\_len$ and $max\_seq\_len$ of the bucket the batch was drawn from (except in merged residual batches). TokenDrop can be synergistically combined with BucketSampler to further equalize the lengths of all sequences in a batch by pruning stopwords from longer sequences in the batch, thereby eliminating the need for padding. To achieve this, we propose defining $TokenDrop\_Rate$ on a sequence-by-sequence basis, rather than having a global $TokenDrop\_Rate$ rate for all sequences. In particular, the number of stopwords to drop in a sequence is computed as $Tokens\_to\_drop(sample) = cardinality(sample) - min\_seq\_len(batch)$. Consequently, all sequences in a batch are pruned to $min\_seq\_len(batch)$ by dropping a random subset of stopwords (Algorithm 3), thereby eliminating padding tokens (except in merged residual batches, where $tokens\_to\_drop(sample)$ may be larger than $num\_stopwords(sample)$ due to larger variance in sequence lengths across merged buckets).

---

**Algorithm 3:** TokenDrop + BucketSampler

**Input** : Input Dataset ($Dataset$), Training epoch ($epoch$), List of stopwords ($Stopwords$)
**Output**: Batches generated from $Dataset$ ($batches$)

1  $batches$ = BucketSampler($Dataset$, $epoch$)
2  **for** $batch$ in $batches$ **do**
3     **for** $sample$ in $batch$ **do**
4        Tokens_to_drop($sample$) = cardinality($sample$) - min_seq_len($batch$)

5  $batches$ = TokenDrop($batches$, $Stopwords$, Tokens_to_drop)
6  **return** $batches$

---

## 3  Experiments and Results

We implement TokenDrop + BucketSampler in PyTorch using Huggingface Transformers (Wolf et al.,

2019). We perform experiments on a NVIDIA RTX 2080 Ti GPU with 11 GB memory, and report results averaged across 10 runs with different random seeds. We perform 3 epochs of fine-tuning for our method and all baselines. The details of all hyperparameters used in our experiments are described in Appendix A. We note that TokenDrop is not used when fine-tuning on CoLA, since the task involves identifying if a given sentence is linguistically acceptable or not, and TokenDrop makes linguistically acceptable sequences unacceptable.

## 3.1 TokenDrop + BucketSampler improves the accuracy and efficiency of fine-tuning

**Classification tasks:** We present results of fine-tuning the popular Roberta (Liu et al., 2019) and Electra (Clark et al., 2020) models on the GLUE (Wang et al.) and SQUADv1.1 (Rajpurkar et al.) datasets in Table 1. We find that fine-tuning with TokenDrop + BucketSampler consistently produces more accurate models compared to conventional fine-tuning with random batches (RandomSampler). In addition, TokenDrop + BucketSampler also reduces the wall-clock fine-tuning time by up to $10.61\times$ compared to conventional fine-tuning, with an average speedup of $5.9\times$ across the 10 GLUE and SQUAD tasks. With RandomSampler, 38.9% of all tokens used for training are padding tokens, which reduces to just 0.2% with TokenDrop + BucketSampler (not exactly 0%, since padding is needed in merged residual batches). We find that the speedup from using TokenDrop + BucketSampler on a given task is dependent on two factors: (1) the sequence length histogram, and (2) the size of the dataset. We provide a detailed analysis of the relationship between the statistics of the fine-tuning dataset and the speedup achieved from using TokenDrop + BucketSampler in Appendix E. We also provide supplementary results on fine-tuning Roberta-Large in Appendix B to demonstrate the benefits of using TokenDrop + BucketSampler for fine-tuning larger models.

**Generation tasks:** We present results of fine-tuning the T5-small seq2seq model (Raffel et al., 2020) on text summarization using the XSum (Narayan et al., 2018) and CNN/DailyMail (Nallapati et al., 2016) datasets in Table 2. We find that fine-tuning with TokenDrop + BucketSampler improves the ROUGE-1 score by up to 0.3 points, while also reducing the wall-clock fine-tuning time by up to $8.62\times$ compared to RandomSampler. We

note that TokenDrop is only applied to input sequences, and bucketing is performed based on input sequence lengths for generation tasks. As a result, padding is still necessary for the target sequences.

**Resilience to minor grammatical errors in inputs:** We find that training with TokenDrop significantly enhances the resilience of fine-tuned models to minor grammatical errors in inputs. For instance, when articles ('a', 'an', 'the') and punctuation marks are removed from the test sequences, the average accuracy on GLUE (except CoLA) and SQUAD drops by 5.2% (Roberta-Base), and the average ROUGE-1 score drops by 3.1 points (T-5 small) for the baseline models. On the other hand, models fine-tuned with TokenDrop incur only 0.3% and 0.06 points drop in average accuracy and ROUGE-1 scores, respectively, thereby demonstrating significantly higher resilience to minor grammatical errors. The enhanced resilience to grammatical errors in models fine-tuned with TokenDrop can also be observed in Fig. 4, where there is negligible loss in accuracy even when 40% of all stopwords in each sequence are randomly chosen and deleted during inference.

## 3.2 TokenDrop + BucketSampler enables accurate and efficient inference

While the primary objective of TokenDrop + BucketSampler is to improve fine-tuning efficiency, we describe how they can also be used to improve the efficiency of both real-time and server-mode inference in the following subsections.

**Real-time inference (batch size = 1):** Real-time inference workloads have strict latency requirements and bursty input rates, and hence, inputs are typically processed as soon as they arrive with a batch size of 1. To reduce the latency of real-time inference, we propose filtering out stopwords in the input text sequence by applying TokenDrop during inference also. Inference-time TokenDrop offers a promising approach for accelerating real-time inference, enabling speedups of 2.2X when all stopwords are pruned ($TokenDrop\_Rate = 1$ in Fig. 4). However, we find that models fine-tuned without TokenDrop suffer from large accuracy drop when TokenDrop is applied at inference time. On the other hand, models trained with TokenDrop exhibit significantly higher resilience to inference-time TokenDrop, enabling $1.5\times$ reduction in inference latency with no loss in accuracy, and $2.2\times$ speedup with $< 1\%$ loss in accuracy (Fig.

Table 1: **Results of fine-tuning Roberta-base and Electra-base on the GLUE and SQUAD v1.1 development sets.** We report F1 score for SQUAD, Matthews correlation for CoLA, Pearson Correlation for STS-B and accuracy for all other tasks. We report only "matched" accuracy for MNLI. Subscripts indicate standard deviation.

| Model | Batching Method | SQUAD | MRPC | STS-B | SST-2 | CoLA | QQP | QNLI | RTE | MNLI | WNLI | Avg |
|---|---|---|---|---|---|---|---|---|---|---|---|---|
| Roberta-Base | RandomSampler | $90.46_{0.2}$ | $88.09_{1.0}$ | $89.85_{0.3}$ | $94.15_{0.5}$ | $59.35_{0.6}$ | $91.36_{0.1}$ | $92.58_{0.1}$ | $69.31_{1.6}$ | $87.56_{0.1}$ | $53.51_{3.5}$ | 81.62 |
| | **TokenDrop+BucketSampler** | $90.59_{0.1}$ | $88.63_{0.8}$ | $90.06_{0.4}$ | $94.33_{0.2}$ | $59.49_{1.1}$ | $91.48_{0.3}$ | $92.64_{0.1}$ | $70.4_{2.8}$ | $87.66_{0.1}$ | $54.57_{2.8}$ | 81.98 |
| Electra-Base | RandomSampler | $90.74_{0.3}$ | $88.28_{0.6}$ | $89.74_{0.4}$ | $95.01_{0.3}$ | $65.48_{0.4}$ | $91.93_{0.1}$ | $92.41_{1.1}$ | $77.07_{0.9}$ | $88.33_{0.6}$ | $54.22_{2.6}$ | 83.32 |
| | **TokenDrop+BucketSampler** | $90.88_{0.2}$ | $88.51_{0.2}$ | $89.79_{0.3}$ | $95.12_{0.3}$ | $66.03_{0.6}$ | $92.11_{0.1}$ | $93.09_{0.3}$ | $78.24_{0.7}$ | $88.74_{0.3}$ | $54.38_{2.8}$ | 83.7 |
| | Average Fine-tuning Speedup | 9.87X | 2.75X | 7.73X | 10.61X | 9.39X | 5.51X | 3.18X | 2.05X | 4.09X | 3.8X | 5.9X |

Table 2: **Results of fine-tuning the T5-small seq2seq model on text summarization.** We report the ROUGE-1 score. Subscripts indicate standard deviation.

| Batching Method | XSum | CNN/DailyMail |
|---|---|---|
| RandomSampler | $32.58_{0.4}$ | $24.51_{0.6}$ |
| **TokenDrop+BucketSampler** | $32.84_{0.3}$ | $24.81_{0.8}$ |
| Speedup | $8.62\times$ | $7.77\times$ |

4). In addition, TokenDrop can be combined with progressive token pruning methods that prune the least-important tokens in each layer based on attention scores (Wang et al., 2021; Goyal et al., 2020), to achieve further gains in efficiency.

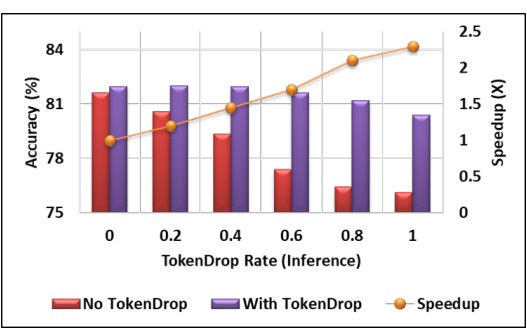

Figure 4: **Accuracy drop and speedups from using TokenDrop during inference with Roberta-base (batch size = 1).** We report the average score on the 9 GLUE tasks and SQUADv1.1. For $0 <$ TokenDrop Rate $< 1$, we randomly prune ($num\_stopwords(sequence) \times TokenDrop\_Rate$) stopwords in each sequence.

**Server-mode inference (batch size $> 1$):** In the server-mode inference setting, inputs arrive simultaneously from several sources. Consequently, inputs are processed in batches, with the goal of maximizing throughput. In the server-mode inference setting, we propose utilizing TokenDrop + BucketSampler to batch the inference queries. Here, we set $BatchCap(bucket) = HWCap(bucket)$ for all buckets to maximize hardware utilization, and hence, throughput. We find that using BucketSampler leads to a $4.5\times$ speedup over random batching. In addition, TokenDrop can also be synergistically combined with BucketSampler at inference time

in models fine-tuned with TokenDrop to achieve $4.9\times$ speedup with no loss in accuracy (Table 3).

### 3.3 Ablation: Breakdown of benefits from the different BucketSampler optimizations

We study the impact of the different BucketSampler optimizations on accuracy and efficiency of fine-tuning in Table 4 and Figure 5. When no optimizations are used, BucketSampler achieves $5.3\times$ speedup, but incurs substantial accuracy drop due to insufficient training convergence. When Residual Batch Merging (RBM) is used, "stray" batches from different buckets are combined to form larger batches. In addition to improving efficiency by reducing hardware under-utilization, RBM also improves accuracy by reducing variability in batch sizes, thereby enabling better convergence with a single learning rate schedule. BatchCap further improves accuracy by using small batch sizes in early epochs, thereby ensuring sufficient numbers of weight updates to achieve training convergence. While BatchCap is necessary for achieving convergence, it leads to small drop in training efficiency due to hardware under-utilization in early epochs. Finally, the use of LRM to dynamically adjust the learning rate for each batch ensures that fine-tuning with BucketSampler results in near-identical training curves (Fig. 5) and hence, accuracy (Table 4) compared to fine-tuning with RandomSampler.

## 4 Related Work

Prior works have developed batching strategies for variable-length inputs to improve the efficiency of LM training. RandomSampler is the most commonly used batching technique, and is the default method in most NLP libraries. RandomSampler randomly selects samples from the training dataset to generate batches, and pads all sequences in a batch to the maximum length of sequences in the batch. Consequently, training with RandomSampler requires substantial padding, and hence, substantial wasted computations. LengthGrouped-Sampler (introduced in Huggingface Transformers

Table 3: **Results of using TokenDrop + BucketSampler during batched inference with Roberta-base.** We report the average score across the 9 GLUE tasks and SQUADv1.1. We assume that all samples in the test dataset arrive simultaneously, and speedup is computed by comparing the wall-clock time taken to infer on all test samples.

| Batching Method (Inference) | Accuracy (Trained without TokenDrop) | Accuracy (Trained with TokenDrop) | Fraction of padding tokens | Speedup |
|---|---|---|---|---|
| RandomSampler | 81.62 | 82.02 | 40.1% | 1X |
| BucketSampler | 81.64 | 81.98 | 9.3% | 4.5X |
| TokenDrop+BucketSampler | 80.21 | 82.01 | 0.3% | 4.9X |

Table 4: **Impact of the different BucketSampler optimizations when fine-tuning Roberta-base.** We report the average score across GLUE and SQUAD.

| Optimizations | Accuracy | Speedup |
|---|---|---|
| None | 76.9 | 5.3× |
| RBM | 78.42 | 5.8× |
| RBM + BatchCap | 81.29 | 5.2× |
| RBM + BatchCap + LRM | 81.64 | 5.2× |

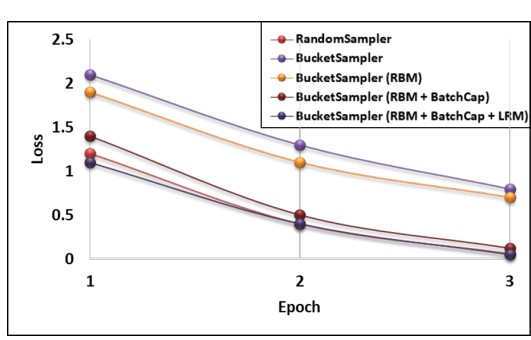

Figure 5: **Average training loss across GLUE and SQUAD when fine-tuning Roberta-base.**

Table 5: **Accuracy and efficiency of fine-tuning Roberta-base with different batching strategies.** Results are averaged across the GLUE tasks and SQUAD.

| Batching Strategy | Accuracy | Speedup |
|---|---|---|
| RandomSampler | 81.62 | 1X |
| LengthGroupedSampler | 81.65 | 3.8X |
| Packing | 80.69 | 4.2X |
| tf.bucket_by_sequence_length | 76.9 | 5.3X |
| BucketSampler | 81.67 | 5.1X |
| **TokenDrop+BucketSampler** | **81.98** | **5.9X** |

(Wolf et al., 2019)) sorts sequences in order of increasing sequence length, and generates batches of adjacent sequences in the sorted list. While LengthGroupedSampler reduces padding, it uses fixed batch sizes, resulting in hardware under-utilization. Packing (Krell et al., 2022) generates batches by concatenating different inputs along the sequence length dimension. However, we find that unlike when training from scratch, the overheads of packing and the additional computations in attention layers (computing irrelevant scores, followed by masking to prevent cross-contamination between different sequences) are not amortized over the small number of fine-tuning epochs. Finally, Tensorflow's tf.data.bucket_by_sequence_length divides the training dataset into buckets based on sequence length, and generates batches only from sequences in the same bucket, similar to BucketSampler. While packing and tf.data.bucket_by_sequence_length support variable batch sizes to maximize hardware utilization, they incur accuracy drop during fine-tuning. In summary, while prior works have demonstrated efficiency gains when training LMs from scratch, the unique characteristics of fine-tuning (small datasets, very few epochs) make these methods ineffective. As a result, TokenDrop + BucketSampler significantly outperforms prior methods in terms of both accuracy and fine-tuning efficiency (Table 5).

## 5 Conclusion

In this work, we presented TokenDrop + Bucket-Sampler for accurate and efficient fine-tuning of LMs. TokenDrop prunes a random subset of stop-words in each sequence in every epoch to reduce overfitting, while BucketSampler creates batches of sequences with similar sequence lengths to reduce the number of padding tokens. We also introduced algorithmic optimizations – Residual Batch Merging (RBM), BatchCap and Learning Rate Modulation (LRM) – to maximize throughput and enable convergence when fine-tuning with BucketSampler. We demonstrated that TokenDrop can be synergistically combined with BucketSampler to drop more tokens from longer sequences in each batch, thereby further reducing the need for padding. In effect, fine-tuning with TokenDrop + BucketSampler produced more accurate models in a shorter time compared to conventional fine-tuning.

## 6 Acknowledgement

This work was supported in part by the Center for the Co-Design of Cognitive Systems (CoCoSys), a JUMP2.0 center sponsored by the Semiconductor Research Corporation (SRC) and DARPA, and in part by SRC under the AIHW program.

# 7 Limitations

(1) TokenDrop is not universally applicable to all fine-tuning tasks. In particular, TokenDrop cannot be used when dropping stopwords can potentially change the labels associated with sequences. For instance, when fine-tuning a LM to identify if a given sequence is grammatically correct or not (as in CoLA), dropping stopwords from sequences will make all sequences grammatically incorrect. (2) BucketSampler is not useful for self-supervised training with the Masked Language Modelling (MLM) objective. When LMs are trained with MLM, the text corpus is divided into fixed-size blocks and fed to the model, thereby resulting in constant input sizes. (3) The use of Batch-Cap in BucketSampler introduces hardware under-utilization in the first epoch of fine-tuning. We find that BatchCap is necessary to achieve convergence when fine-tuning with TokenDrop + BucketSampler with the same set of hyperparameters as RandomSampler. However, it is possible that the use of advanced optimizers and/or learning rate schedules can enable convergence without the need for Batch-Cap, thereby further increasing efficiency. We plan to explore this as part of future work.

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

# A   List of hyperparameters

The hyperparameters used in our experiments are described in Table 6. We reduce the dropout rate when training with TokenDrop + BucketSampler compared to training with RandomSampler in order to account for the regularization effect of TokenDrop. In particular, we set the dropout rate to 0 on small datasets ($<$10K training samples) to enable convergence within 3 epochs. For larger datasets with $>$10K training samples, we reduce the dropout rate to 0.025, since sufficient weight updates are performed to enable convergence even when small amount of dropout is used, and the additional regularization from dropout leads to higher test accuracy.

**Choice of buckets:** The parameters $min\_seq\_len$ and $max\_seq\_len$ of a bucket determine the sequence length range of all samples in the bucket. When TokenDrop is not used, we simply define buckets in length increments of 1 i.e., bucket 1 will have sequences of length 1, bucket 2 will have sequences of length 2, and so on. Therefore, the number of buckets will be equal to the $max\_seq\_len$ of the dataset, and the $[min\_seq\_len, max\_seq\_len)$ of bucket $i$ will be $[i,i)$. This will eliminate the need for padding in non-residual batches (since all sequences in a bucket will have the same sequence length), and RBM will keep hardware utilization as high as possible while minimizing padding in residual batches. When TokenDrop is used, the sequence length spread ($max\_seq\_len$ - $min\_seq\_len$ + 1) of each bucket must be $>$1 to enable token dropping in non-residual batches also. However, if the sequence length spread of each bucket is too large, then it is highly likely that $Tokens\_to\_drop(sample)$ will be larger than $number\_of\_stopwords(sample)$ for most samples. This has two drawbacks: (1) it necessitates the use of padding, and (2) all stopwords in many samples will be dropped in every epoch, thereby reducing data diversity between epochs and leading to weaker regularization. In our experiments, we find that defining buckets in increments of 5, where bucket 1 has ($min\_seq\_len$, $max\_seq\_len$) = [0,5), bucket 2 has ($min\_seq\_len$, $max\_seq\_len$) = [5,10) and so on, provides good trade-off between minimizing padding and ensuring sufficient regularization for high test accuracy.

**List of stopwords used for TokenDrop:** ['i', 'me', 'my', 'myself', 'we', 'our', 'ours', 'ourselves', 'you', "you're", "you've", "you'll", "you'd", 'your', 'yours', 'yourself', 'yourselves', 'he', 'him', 'his', 'himself', 'she', "she's", 'her', 'hers', 'herself', 'it', "it's", 'its', 'itself', 'they', 'them', 'their', 'theirs',

Table 6: **List of hyperparameters used in our experiments.** We use the same set of hyperparameters for all GLUE and SQUAD tasks. We also use the same set of hyperparameters when fine-tuning both Roberta-base and Electra-base. HWCap is experimentally determined on a NVIDIA RTX 2080Ti GPU with 11GB memory.

| Hyperparameter | Value (RandomSampler) | Value (TokenDrop + BucketSampler) |
|---|---|---|
| Batch Size | 64 | Variable |
| Learning Rate | Best among $\{3, 4, 5\} * e^{-5}$ | Variable |
| Dropout Rate | 0.1 | 0 for {MRPC, STS-B, CoLA, RTE, WNLI}, 0.025 for {SST-2, QQP, QNL, MNLI, SQUAD} |
| Buckets (min_seq_len, max_seq_len, HWCap) | N/A | Defined in increments of 5, i.e., (0,5,2525), (5,10,1250), (10,15,800), ..., (120,125,64), (125,128,64) |
| BatchCap: (scaling_factor, base_batch_size) | N/A | (2, 64); base_batch_size = Batch Size used for RandomSampler |
| LRM: (base_learning_rate, base_batch_size) | N/A | (Learning Rate, Batch Size) used for RandomSampler i.e., (best among $\{3, 4, 5\} * e^{-5}$, 64) |

'themselves', 'what', 'which', 'who', 'whom', 'this', 'that', "that'll", 'these', 'those', 'am', 'is', 'are', 'was', 'were', 'be', 'been', 'being', 'have', 'has', 'had', 'having', 'do', 'does', 'did', 'doing', 'a', 'an', 'the', 'and', 'but', 'if', 'or', 'because', 'as', 'until', 'while', 'of', 'at', 'by', 'for', 'with', 'about', 'against', 'between', 'into', 'through', 'during', 'before', 'after', 'above', 'below', 'to', 'from', 'up', 'down', 'in', 'out', 'on', 'off', 'over', 'under', 'again', 'further', 'then', 'once', 'here', 'there', 'when', 'where', 'why', 'how', 'all', 'any', 'both', 'each', 'few', 'more', 'most', 'other', 'some', 'such', 'only', 'own', 'so', 'than', 'too', 'very', 'can', 'will', 'just', 'should', "should've", 'now', '-', ',', '.', ';', '...', '(', ')']. We remove all pronouns from the stopword list on tasks where they are required (such as WNLI, where the task is to select the referent of a given pronoun from a list of choices).

## B  Results of fine-tuning Roberta-Large

We present results of fine-tuning Roberta-Large (Liu et al., 2019) on GLUE (Wang et al.) and SQUADv1.1 (Rajpurkar et al.) in Table 7. We find that fine-tuning with TokenDrop + BucketSampler produces models that are up to 1.96% more accurate, while also reducing the wall-clock fine-tuning time by up to 10.12× compared to fine-tuning with RandomSampler. The average accuracy gain from using TokenDrop + BucketSampler on Roberta-Large is more than the gain on Roberta-Base, since overfitting is a bigger problem on larger models. In particular, Roberta-Large has approximately 3× the number of parameters as Roberta-Base, making it more susceptible to memorizing the limited training samples during fine-tuning. On the other hand, the fine-tuning speedup from using TokenDrop + BucketSampler on Roberta-Large is similar to the speedup on Roberta-Base, since speedup depends on the dataset statistics rather than the model architecture.

## C  Ablation: Necessity of dropping a random subset of stopwords with TokenDrop

We demonstrate the need for dropping a random subset of stopwords from sentences in Table 8. When all tokens are considered for random dropping, fine-tuning does not converge in the small number of epochs typically used, since dropping important tokens changes the meaning of the input sequences. On the other hand, dropping only a random subset of stopwords does not alter the meaning of sentences, thereby enabling convergence while also preventing overfitting. We also find that dropping all stopwords in each sequence in every epoch leads to loss in accuracy. This is because there is no longer a regularization effect, since the same sequences (with all stopwords dropped) are presented to the model in every epoch, and the pruning of stopwords makes sequences shorter, and potentially easier for the model to memorize (Table 8).

## D  Reducing overfitting with TokenDrop

We analyze the effectiveness of TokenDrop in reducing overfitting when fine-tuning on three datasets of different sizes – small (RTE, with 2.5K training samples), medium (SST-2, with 67K training samples), and large (MNLI, with 393K training samples) – in Fig. 6. We find that the training loss decreases more gradually when TokenDrop is used, since different data samples are presented in each epoch. As a result, the model does not simply memorize the training data within a few epochs, leading to better generalization performance.

Table 7: **Results of fine-tuning Roberta-Large on the GLUE and SQUAD v1.1 development sets.** We report F1 score for SQUAD, Matthews correlation for CoLA, Pearson Correlation for STS-B and accuracy for all other tasks. We report only "matched" accuracy for MNLI. Subscripts indicate standard deviation.

| Batching Method | SQUAD | MRPC | STS-B | SST-2 | CoLA | QQP | QNLI | RTE | MNLI | WNLI | Avg |
|---|---|---|---|---|---|---|---|---|---|---|---|
| RandomSampler | $91.33_{0.3}$ | $89.71_{0.8}$ | $91.65_{0.7}$ | $96.28_{1.0}$ | $65.49_{1.4}$ | $92.38_{0.1}$ | $93.72_{0.1}$ | $87.1_{2.1}$ | $89.58_{0.1}$ | $72.12_{4.2}$ | 86.94 |
| **TokenDrop+BucketSampler** | $\mathbf{91.68_{0.2}}$ | $\mathbf{90.56_{0.5}}$ | $\mathbf{91.82_{0.7}}$ | $\mathbf{96.51_{0.6}}$ | $\mathbf{65.54_{1.3}}$ | $\mathbf{92.74_{0.3}}$ | $\mathbf{93.99_{0.2}}$ | $\mathbf{88.34_{2.6}}$ | $\mathbf{89.88_{0.1}}$ | $\mathbf{74.08_{3.7}}$ | **81.98** |
| Speedup | 9.46X | 2.91X | 7.28X | 10.12X | 8.96X | 5.6X | 3.14X | 2.02X | 4.19X | 3.92X | 5.76X |

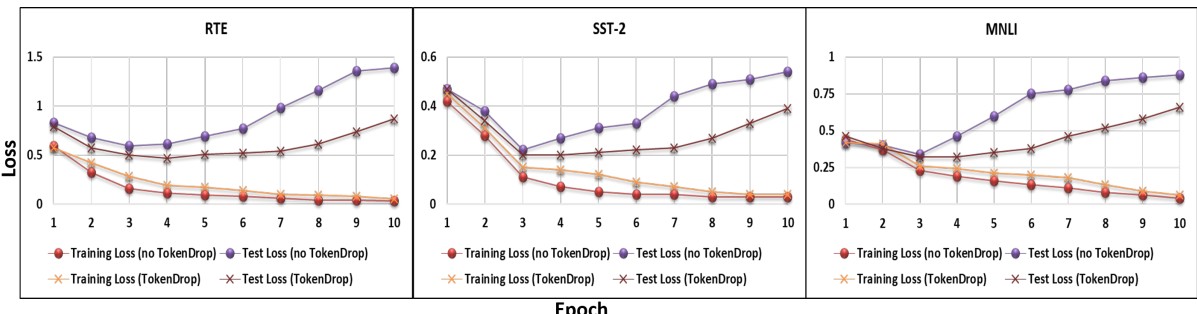

Figure 6: **Training curves obtained from fine-tuning Roberta-base.** We report loss averaged across 10 random seeds. Dropout rate = 0.1 is used for all datasets when fine-tuning without TokenDrop. When fine-tuning with TokenDrop, Dropout Rate = {0 for RTE, 0.025 for SST-2 and MNLI}, as listed in Table 6.

Table 8: **Evaluation of regularization strategies.** We report the average score across GLUE and SQUAD.

| Regularization Method | Accuracy |
|---|---|
| Dropout (Dropout rate = 0.1) | 81.62 |
| Drop any random token (Drop rate = 0.3) | 78.44 |
| **TokenDrop: Drop only stopwords (TokenDrop rate = 0.3)** | **81.98** |
| Drop all stopwords (TokenDrop rate = 1.0) | 79.97 |

# E    Factors impacting speedup from using TokenDrop + BucketSampler on a given task

We find that the size of the fine-tuning dataset and the sequence length spread play key roles in determining the speedup achieved when using TokenDrop + BucketSampler on a given task. We analyze these relationships in the following subsections.

## E.1    Dataset size

Both TokenDrop and BucketSampler add small overheads at fine-tuning time. When TokenDrop is used, stopwords are first identified in each sequence. Then, a random subset of stopwords are pruned from each sequence in every epoch. When BucketSampler is used, the training dataset is first divided into buckets. Then, batches are randomly generated from each bucket in every epoch, and residual batches from different buckets are merged. Finally, the generated batches are shuffled to randomly order batches from different buckets. As a result, once the buckets are generated at the start

of fine-tuning, the other steps of BucketSampler are very similar to those in RandomSampler (with the exception of RBM, which accounts for a very small fraction of the runtime), and hence, the overheads are negligible. We analyze the overheads of TokenDrop + BucketSampler on three fine-tuning datasets of different sizes – small (RTE, with 2.5K training samples), medium (SST-2, with 67K training samples), and large (MNLI, with 393K training samples) – in Figure 7. We find that the most time-consuming parts of TokenDrop (identifying all stopwords in each sequence) and BucketSampler (dividing the dataset into buckets) are performed only once at the start of fine-tuning. On the other hand, the operations performed in each epoch account for only a small fraction of the total runtime. Consequently, the overheads of TokenDrop and BucketSampler are better amortized over the course of fine-tuning when training on larger datasets, leading to higher speedups.

**Further accelerating hyperparameter search during fine-tuning.** When hyperparameter tuning is necessary, it is sufficient to split the dataset into buckets and perform stopword identification only once, during the first epoch of the first fine-tuning run. They can then be re-used for subsequent runs with different hyperparameters. As a result, if we compare the wall-clock time taken to perform fine-tuning with 10 different random seeds, TokenDrop + BucketSampler achieves an average

speedup of 6.8× over RandomSampler across the 9 GLUE tasks and SQUAD. We also note that all speedups reported in Section 3 are not computed this way. Instead, it is assumed that both steps (stopword identification and bucket generation) are performed in every fine-tuning run, even when results are averaged across multiple random seeds.

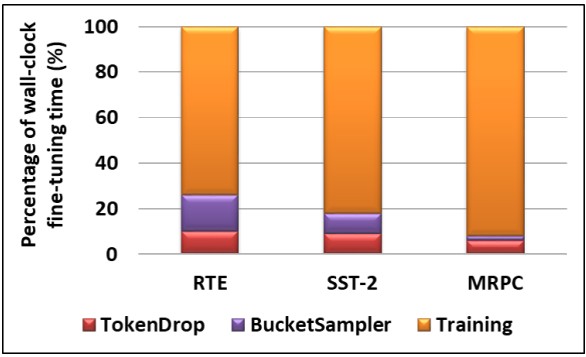

Figure 7: **Overheads of TokenDrop + BucketSampler.** Times are measured on a NVIDIA RTX 2080 Ti GPU with 11 GB memory.

### E.2 Sequence length spread of a dataset

We quantify the sequence length spread of a dataset using two parameters: $L_{avg}$ and $L_K$. $L_{avg}$ is the average sequence length of all sequences in the dataset, while $L_K$ is the maximum possible sequence length such that at least $K\%$ of all sequences in the dataset are longer than $L_K$. When $K = 87.5\%$ and assuming a batch size of 64 ($HWCap = 64$ when $max\_seq\_len = 128$ on a NVIDIA RTX 2080 Ti GPU), the probability of each batch having at least one sequence with length $> L_K = (1 - (87.5/100)^{64}) = 0.9999$ when batching with RandomSampler. As a result, it is expected that each batch will be padded to at least $L_{87.5}$. Consequently, $padding\_fraction\_est = ((L_{87.5} - L_{avg})/L_{87.5})$ is a conservative estimate of the fraction of all tokens that are expected to be padding tokens when batching with RandomSampler. We observe that $padding\_fraction\_est$ has a direct correlation with speedup achieved when using TokenDrop + BucketSampler (Fig. 8). We observe some outliers when the datasets are very small, as in the case of RTE (2.5k training samples) and WNLI (634 training samples), since the overheads of TokenDrop + BucketSampler account for a larger fraction of the wall-clock fine-tuning time (see Fig. 7). We achieve maximum speedup on SST-2, a relatively large dataset (67K training samples) with high $padding\_fraction\_est$ (nearly

50% of all tokens are expected to be padding tokens with RandomSampler). In addition, SST-2 has $L_{avg} = 14$, and hence, the majority of batches can be processed with batch sizes $>800$ ($HWCap = 800$ for the bucket with $max\_seq\_len = 15$), leading to large speedups over RandomSampler (where the batch size of all batches is determined by the longest sequence in the dataset, leading to hardware under-utilization when processing batches with short sequences only).

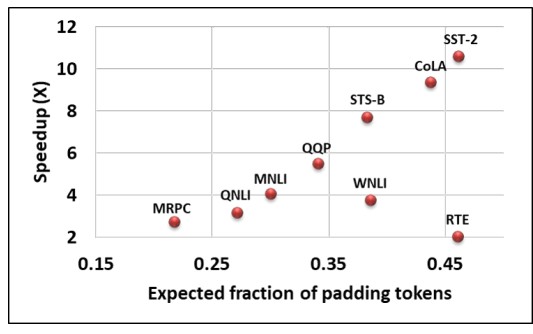

Figure 8: **Impact of sequence length spread of a dataset on fine-tuning speedup achieved using To-kenDrop + BucketSampler.**