# OpenReview forum: "TokenDrop + BucketSampler: Towards Efficient Padding-free Fine-tuning of Language Models"
_EMNLP/2023/Conference — EMNLP 2023 Findings_

### Official Review · Reviewer_4fLH · 2023-07-23

**Soundness:** 3

**Excitement:**

3: Ambivalent: It has merits (e.g., it reports state-of-the-art results, the idea is nice), but there are key weaknesses (e.g., it describes incremental work), and it can significantly benefit from another round of revision. However, I won't object to accepting it if my co-reviewers champion it.

**Missing References:**

- Bucketing was a technique that was (perhaps introduced) and used widely when RNNs and LSTMs became very popular around 2015/6. There are no references to those earlier works.

**Paper Topic And Main Contributions:**

The paper introduces two techniques or tricks to utilities the hardware for training large language models, while avoiding degrading of accuracy. These two methods are:
- BucketSampler: they create batches based on the size of the sequences. Longer sequences are grouped together and shorter ones are grouped together.
- Token Drop: as a form of regulariser and also to reduce padding, the authors introduce TokenDrop. By dropping tokens, at a higher rate for longer sequences, they reduce the need for padding and therefore, making the training faster and more efficient.
- They also use a few more tricks to avoid large batches (batch cap) or small batches (residual batch merging)

Their results on GLUE and SQUAD dataset, using Roberta model only, shows that using all the techniques introduce, there could be a speed up gain of up to x10, while avoiding performance drop. The techniques are more effective when a dataset has sequences of a varied length, especially higher number of short sequences.

**Questions For The Authors:**

1. At line 8, authors mention "which is expected to be performed much more broadly albeit at lower computational cost.", what do they mean by more broadly?
2. In line 442, authors mention that "the datasets with larger numbers of short sequences benefit more from BucketSampler".  Why that is the case? My intuition is that when the distribution of the length is spread, with sequences of varying lengths, we would benefit more from this technique. Would it be possible to see where the higher percentage of shorter sequences has a correlation with speed up gains?
3. In section 2.3, authors mention min_seq_len and max_seq_len. How are these selected?
4. How did you choose the dynamic learning rate for a batch in line 379? What was the intuition, did you try different formulas?
5. In line 316, O(n) has been used. Why do we need the complexity notation? Why not just _n_?
6. In line 258 authors mention "HWCap is experimentally determined by profiling on-chip memory usage for different sequence lengths.". Does this mean that for every hardware or model, one has to do manual inspection. If so, is this a limitation on using the method out of the box?
7. Authors claim (line 180) that "we find that dropout is ineffective at preventing overfitting during fine-tuning (Fig. 1). The Figure depicts only RoBERTa base model on RTE task. Can we make such a broad claim based on one setting?

**Reasons To Accept:**

The techniques authors have introduced and their experiments show that fine-tuning and inference of ReBERTa can be made faster by up to 10x on some datasets, specifically GLUE and SQUAD.

While authors have experimented with RoBERTa, this method cab be applicable to other transformer models.
The authors have carried out an ablation study to show which techniques can lead to speed up or prevent performance drop.


**Reasons To Reject:**

- While authors mention LengthGroupedSampler in the related work section, there is no comparison in the experiments.
- I would like to have seen more analysis of the results. for instance, it will be nice to see the relationship between the spread of length across a dataset and the speed up gain. In Table 1, we see that speed up gain is different for different dataset but it will be more helpful to see why that is.
- The authors have only experimented with RoBERTa base. It will be good to include experiments with seq2seq models and also slightly larger models, for instance RoBERTa large, to see how different architectures or size affect the speed up or performance drop.

**Reproducibility:**

4: Could mostly reproduce the results, but there may be some variation because of sample variance or minor variations in their interpretation of the protocol or method.

**Reviewer Confidence:**

4: Quite sure. I tried to check the important points carefully. It's unlikely, though conceivable, that I missed something that should affect my ratings.

**Typos Grammar Style And Presentation Improvements:**

in the second page, when authors are listing their contributions, there are outlined in 4 points. However, those 4 points are repetition of the TokenDrop and BucketSampler. I suggest authors make it more concise and reduce the contribution two two points 1. TokenDrop 2. BucketSampler. Authors can add another bullet point for the extra techniques they introduced such as dynamic learning rate or residual batch merging.

---

> ### Author Rebuttal · Authors · 2023-08-29
>
> We thank the reviewer for the detailed feedback. We address the main questions and comments below, including a new comparison and new results on larger models.
>
> **Re: While authors mention LengthGroupedSampler in the related work section, there is no comparison in the experiments.**
>
> We compare TokenDrop+BucketSampler with LengthGroupedSampler in Table 4 (page 8 in our paper) in terms of both accuracy and efficiency. In summary, we find that TokenDrop+BucketSampler achieves ~0.3% higher accuracy and 1.6X speedup over LengthGroupedSampler averaged across the GLUE tasks and SQUAD with Roberta-Base. We can also report the speedup over LengthGroupedSampler for each task (similar to Table 1) in the final paper. We chose RandomSampler as baseline for the primary results since it is by far the most widely used method in practice for fine-tuning LLMs.
>
> **Re: The authors have only experimented with RoBERTa base. It will be good to include experiments with seq2seq models and also slightly larger models, for instance RoBERTa large, to see how different architectures or size affect the speed up or performance drop.**
>
> In addition to Roberta-Base, we also provide results on Electra-Base in Table 1 (page 7 in our paper).
>
> **Larger models** – Following the reviewer’s suggestion, we performed experiments with Roberta-Large on GLUE and SQUAD. When RandomSampler is used, we obtain an average score of 86.94 across the 10 tasks (averaged across 10 random seeds). With TokenDrop+BucketSampler, the average score increases to 87.51. Therefore, TokenDrop+BucketSampler leads to 0.57% accuracy gain on Roberta-Large. The accuracy gain on Roberta-Large is more than the gain on Roberta-Base, since overfitting is a bigger problem on larger models. Fine-tuning with TokenDrop+BucketSampler is also 6X faster than fine-tuning with RandomSampler. The speedup on Roberta-Large is very similar to the speedup on Roberta-Base, since speedups depend on the dataset statistics rather than the models (max batch size with RandomSampler and HWCap for the buckets are affected by similar amounts when moving to larger models, so speedups remain approximately constant across model sizes). We will add task-wise results on Roberta-Large to Table 1 in the final paper.
>
> **seq2seq models** – We experiment by using TokenDrop+BucketSampler to fine-tune the T5-small seq2seq model on xsum, a text summarization task. We achieve an average ROGUE-1 score of 32.84 when fine-tuning with TokenDrop+BucketSampler, which is 0.26 points higher than the score obtained using RandomSampler. In addition, fine-tuning with TokenDrop+BucketSampler is 8.62X faster than RandomSampler.
>
> We also find that fine-tuning with TokenDrop **significantly increases the resilience of LLMs to grammatical errors** in text sequences during inference. LLMs trained with RandomSampler are adversely impacted by even minor grammatical errors. For instance, if only articles (“a”, “an”, “the”) were dropped from the test sequences, the average ROUGE-1 score drops from 32.58 to 29.98. On the other hand, when LLMs are trained with TokenDrop, the drop in ROUGE-1 score is only 0.05 (32.84 -> 32.79) when articles are dropped, indicating that they are resilient to minor grammatical errors. Since conventional fine-tuning (RandomSampler) uses only grammatically perfect sequences to train the model, high accuracy is obtained only on grammatically perfect test sequences. This can also be seen in Fig 3 (page 7 in our paper), where dropping only a small subset of stopwords that do not change the meaning of sequences leads to a large drop in accuracy on models trained with RandomSampler. This is important because the assumption of seeing only grammatically correct sequences during inference does not always hold in real-world scenarios, especially since non-native speakers also use these tools.
>
> **Re: At line 8, authors mention "which is expected to be performed much more broadly albeit at lower computational cost.", what do they mean by more broadly?**
>
> Currently, pre-training of LLMs can only be performed by those with access to large GPU/ TPU clusters. As a result, fine-tuning pre-trained LLMs is the primary way of leveraging LLMs for the vast majority of users. This is what we meant by “more broadly”. Pre-trained LLMs that are publicly available (such as BERT, OPT, etc.) are fine-tuned millions of times by large numbers of users on a diverse range of downstream tasks. We will make this clearer in the final paper.
>
> **Re: In line 442, authors mention that "the datasets with larger numbers of short sequences benefit more from BucketSampler". Why that is the case? It will be nice to see the relationship between the spread of length across a dataset and the speed up gain.**
>
> That line was incomplete. We thank the reviewer for pointing this out, and we will fix it in the final paper. Potential for speedup is maximum when a majority of sequences in the dataset are short, but there are a substantial number of long sequences also. For example, in the SST-2 dataset, approximately 80% of all sequences have length <20, but approximately 10% of all sequences have length >35. As a result, when using RandomSampler with a batch size of 64, there is a >99% chance that each batch will have at least one sequence with length >35, leading to all samples in the batch being padded to >35 tokens. In contrast, when using TokenDrop+BucketSampler, the majority of batches can be processed with length <20, leading to large speedups over RandomSampler. This can be seen in Table 1, where speedup is maximum on SST-2. We will add a detailed analysis in the final paper about the relationship between the sequence length spread and size of a dataset, and the speedup achieved.
>
> **Re: In section 2.3, authors mention min_seq_len and max_seq_len. How are these selected?**
>
> min_seq_len and max_seq_len of a bucket determine the sequence length range of all samples in the bucket. When TokenDrop is not used, we simply define buckets in length increments of 1 i.e., bucket 1 will have sequences of length 1, bucket 2 will have sequences of length 2, and so on. Therefore, the number of buckets will be the max_seq_len of the dataset, and the [min_seq_len, max_seq_len) of bucket i will be [i,i). This will eliminate the need for padding in non-residual batches (since all sequences in a bucket will have the same sequence length), and RBM will keep hardware utilization as high as possible while minimizing padding in residual batches. When TokenDrop is used, the sequence length spread (max_seq_len - min_seq_len + 1) of each bucket must be >1 to enable token dropping in non-residual batches also. However, if the sequence length spread of each bucket is too large, then it is highly likely that Tokens_to_drop(sample) will be larger than number_of_stopwords(sample) for most samples (Algorithm 3, line 4, page 6). This has two drawbacks: (1) it necessitates the use of padding, and (2) all stopwords in many samples will be dropped in every epoch, thereby reducing data diversity between epochs and leading to weaker regularization. In our experiments, we find that defining buckets in increments of 5, where bucket 1 has (min_seq_len, max_seq_len) = [0,5), bucket 2 has (min_seq_len, max_seq_len) = [5,10) and so on, provides good trade-off between minimizing padding and ensuring sufficient regularization for high validation accuracy. We thank the reviewer for pointing out this missing detail, and we will add an additional subsection in Appendix A describing how hyperparameters were chosen to complement the list of hyperparameters in Table 5.
>
> **Re: How did you choose the dynamic learning rate for a batch in line 379? What was the intuition, did you try different formulas?**
>
> Scaling learning rate by a factor of sqrt(k) when the batch size is scaled by a factor of k is a popular method for parallelizing training of deep neural networks on large GPU clusters (https://arxiv.org/abs/1404.5997). In particular, when batch size is changed to k * batch_size, the learning rate is changed to learning_rate * sqrt(k). Prior works have used this square root scaling for fixed batch sizes (i.e., batch size is determined at the start of training based on the number of GPUs available) to find the optimal learning rate for a given batch size when the optimal learning rate is known for a different batch size. In this work, we find that this formulation works well for variable batch sizes also (i.e., changing learning rate for each batch based on the size of the batch). We thank the reviewer for pointing out this missing motivation, and we will add this to the final paper along with the relevant references.
>
> **Re: In line 316, O(n) has been used. Why do we need the complexity notation? Why not just n?**
>
> The reviewer is correct. There is no need for the complexity notation. We will change it to just n in the final paper.
>
> **Re: In line 258 authors mention "HWCap is experimentally determined by profiling on-chip memory usage for different sequence lengths.". Does this mean that for every hardware or model, one has to do manual inspection. If so, is this a limitation on using the method out of the box?**
>
> We automate this entire process through a simple script that tracks the peak memory usage of a single forward + backward pass for a given model and sequence length with a random base batch size. We then increase or decrease the batch size depending on the ratio of peak memory usage to hardware memory capacity, and this process is repeated till the peak memory usage is less than but close to the hardware memory capacity. These steps are then repeated for different sequence lengths under consideration. Since this process is highly efficient and takes less than a minute for all models in our experiments, we do not believe that this will be a limitation on using our methods out of the box.
>
> **Re: Authors claim (line 180) that "we find that dropout is ineffective at preventing overfitting during fine-tuning (Fig. 1). The Figure depicts only RoBERTa base model on RTE task. Can we make such a broad claim based on one setting?**
>
> We observe similar trends on other tasks also, with both Roberta-Base and Electra-Base. We also observe that the overfitting problem is even more severe on Roberta-Large. We will add more curves when fine-tuning on datasets of different sizes in the final paper. We chose RTE to demonstrate how severe the overfitting problem can be during fine-tuning even with base models, and to motivate the need for stronger regularization.
>
> **Re: Missing References, Typos Grammar Style And Presentation Improvements**
>
> We thank the reviewer for the suggestions and for pointing out a source of relevant references. We will incorporate them in the final paper.

---

### Official Review · Reviewer_4LK5 · 2023-08-02

**Soundness:** 4

**Excitement:**

4: Strong: This paper deepens the understanding of some phenomenon or lowers the barriers to an existing research direction.

**Missing References:**

not extremely well-versed with the literature so not sure of this if any

**Paper Topic And Main Contributions:**

This paper proposes a framework to improve efficiency during fine-tuning of language models, without harming the accuracy. The efficiency speedup is made possible by BucketSampler, which creates batches of samples having similar sequence lengths. BucketSampler has three additional components, the first two of which are necessary to achieve convergence: a) modulating the learning rate based on the batch size; b) having a cap on the maximum batch size (which can grow large for batches that are composed of small sequences) depending on the training epoch to prevent sparse updates; c) a way to merge residual samples from multiple buckets. TokenDrop is a method that drops tokens (specifically, stopwords only) from sequences, which serves the dual purpose of acting as a regularizer and creating batches of equal length (once they have been bucket-ed). Their framework is tested on roberta-base and electra-base for all the GLUE tasks and SQuAD. They achieve an average speedup of 5.9x with slightly better (~0.3% gain) accuracies.

**Questions For The Authors:**

A) The title and the build-up to the experimental section is slightly misleading. The title says "large language models" but the term "large" is very subjective. The introduction mentions "LLMs are characterized by large model sizes (exceeding 1 Trillion parameters), and are pre-trained on very large text corpora", but experiments are conducted on base models (~110 million parameters). While it is definitely not an issue if the experiments are done on base models, it would benefit the reader if the wording aligns with the experiments, and removing the word "large" throughout would help achieve that.

B) L377-381: How does this formulation of LRM come into place? Is it borrowed from previous literature or motivated due to other reasons?  Motivating its formulation might help add clarity to the reader.

C) Since there are multiple components involved in the framework, laying out its scope in terms of tasks and models it would benefit would be really helpful for its usability in the real world.

**Reasons To Accept:**

A) Writing: The paper is easy-to-understand and the framework is detailed. The motivation is clear and the shortcomings of past works are well laid-out.

B) Code and Soundness: The code is released and results reported are averaged across 10 seeds. The framework is compared to several past methods and lay out the advantages over them.

C) Ablation: Each piece of the framework and its contribution is studied in Table 3.

D) The efficiency speedups are impressive, all while maintaining accuracy.

**Reasons To Reject:**

A) Scope and Generalizability: The major weakness in the framework's design is its limited scope to multiple tasks and languages. With regards to tasks, it is not applicable to a few classification- and all generation-tasks. Amongst classification tasks, TokenDrop cannot be used for COLA (which judges grammaticality), and generation tasks (like machine translation or summarization) wouldn't allow for ungrammatical outputs either (although the paper does not mention generation tasks but clarification on the scope would be beneficial). In general, the method presents a mismatch between pre-training and finetuning, which can be problematic in a few settings like zero-shot cross-lingual transfer in multilingual models. If a multilingual model is fine-tuned in English with stopwords dropped it might affect cross-lingual generalization. One would also require knowing stopwords apriori if finetuning on different languages.
With regards to models, the framework is tested on base models (110M params), while being motivated as an efficient LLM fine-tuning framework. I do not see the experimental setup of testing only base models to be problematic, but its framing is misleading (elaborated in the questions section). However, I bring up this point here because the method involves optimization tricks like learning rate modulation and BatchCap, whose generalizability to larger models is uncertain. If instead of electra-base for example, the paper would include roberta-base and roberta-large (300-350M params?), it would alleviate some of these concerns.

**Reproducibility:**

5: Could easily reproduce the results.

**Reviewer Confidence:**

4: Quite sure. I tried to check the important points carefully. It's unlikely, though conceivable, that I missed something that should affect my ratings.

**Typos Grammar Style And Presentation Improvements:**

A) The introduction and Section 2 (L158-171) while clear, seems repetitive. For example, the method and its motivation is described multiple times (L010-L32, L076-L129, L141-L151, L158-L171). This can be cut down to two overall places of description instead of 4, one in the abstract and one the introduction. The saved space can be used to create a visualization for Section 2.2 which describes BucketSampler and the three components with a toy example?

---

> ### Author Rebuttal · Authors · 2023-08-29
>
> We thank the reviewer for the thoughtful feedback. We address the main questions and comments below and provide new results on larger models.
>
> **Re: Applicability to generation tasks**
>
> While the reviewer is correct that generalization tasks do not allow ungrammatical outputs, TokenDrop is applied only on the input text sequences during fine-tuning, and never on the output sequences. As a result, TokenDrop can be used for generalization tasks, as long as applying TokenDrop on the input does not change the label (like in the case of CoLA, where grammatically correct sequences become incorrect when TokenDrop is used). We validate this by using TokenDrop+BucketSampler to fine-tune the T5-small seq2seq model on xsum, a text summarization task. We achieve an average ROGUE-1 score of 32.84 when fine-tuning with TokenDrop+BucketSampler, which is 0.26 points higher than the score obtained using RandomSampler. In addition, fine-tuning with TokenDrop+BucketSampler is 8.62X faster than RandomSampler.
>
> We also find that fine-tuning with TokenDrop **significantly increases the resilience of LLMs to grammatical errors** in text sequences during inference. LLMs trained with RandomSampler are adversely impacted by even minor grammatical errors. For instance, if only articles (“a”, “an”, “the”) were dropped from the test inputs, the average ROUGE-1 score drops from 32.58 to 29.98. On the other hand, when LLMs are trained with TokenDrop, the drop in ROUGE-1 score is only 0.05 (32.84 -> 32.79) when articles are dropped, indicating that they are resilient to minor grammatical errors. Since conventional fine-tuning (RandomSampler) uses only grammatically perfect sequences to train the model, high accuracy is obtained only on grammatically perfect test sequences. This can also be seen in Fig 3 (page 7 in our paper), where dropping only a small subset of stopwords that do not change the meaning of sequences leads to a large drop in accuracy on models trained with RandomSampler. This is important because the assumption of seeing only grammatically correct inputs during inference does not always hold in real-world scenarios, especially since non-native speakers also use these tools.
>
> **Re: Results on larger models**
>
> Following the reviewer’s suggestion, we experiment with fine-tuning Roberta-Large on GLUE and SQUAD. When RandomSampler is used, we obtain an average score of 86.94 across the 10 tasks (averaged across 10 random seeds). With TokenDrop+BucketSampler, the average score increases to 87.51. Therefore, TokenDrop+BucketSampler leads to 0.57% accuracy gain on Roberta-Large. The accuracy gain on Roberta-Large is more than the gain on Roberta-Base, since overfitting is a bigger problem on larger models. Fine-tuning with TokenDrop+BucketSampler is also 6X faster than fine-tuning with RandomSampler. The speedup on Roberta-Large is very similar to the speedup on Roberta-Base, since speedups depend on the dataset statistics rather than the models (max batch size with RandomSampler and HWCap for the buckets are affected by similar amounts when moving to larger models, so speedups remain approximately constant across model sizes). We will add task-wise results on Roberta-Large to Table 1 in the final paper.
>
> **Re: Applicability to different languages, and the need to know stopwords apriori**
>
> The reviewer is correct about needing to know the stopwords apriori for different languages.  Lists of stopwords in most major languages are publicly available (https://github.com/6/stopwords-json, https://advertools.readthedocs.io/en/master/advertools.stopwords.html etc.). However, we acknowledge that these lists may not be available for low-resource and understudied languages. We will add this to the limitations section in the final paper.
>
> **Cross-lingual generalizability** – Intuitively, we do not expect TokenDrop to affect cross-lingual generalizability since (a) it only drops words that do not contribute to the meaning of the sentence, and (b) only a random subset of stopwords are dropped in each iteration i.e., not all stopwords are dropped from every sequence, and a different set of stopwords are dropped from a given sequence in each epoch. However, we are unable to provide empirical evidence for this in the short rebuttal period. We will either empirically show that TokenDrop does not affect cross-lingual transfer in the final paper, or call it out in the limitations section.
>
> **Re: The title and the build-up to the experimental section is slightly misleading**
>
> We used the term LLM to refer to a particular class of models – pre-trained foundation models that are subsequently fine-tuned for different NLP tasks – rather than models that are enormous. However, after re-reading the paper in light of the reviewer’s comments, we agree that the title and intro may be slightly misleading. We will remove references to “large” in the final version, and place more emphasis on the class of models we are targeting (foundation models or language models)
>
> **Re: How does this formulation of LRM come into place? Is it borrowed from previous literature or motivated due to other reasons? Motivating its formulation might help add clarity to the reader.**
>
> Scaling learning rate by a factor of sqrt(k) when the batch size is scaled by a factor of k is a popular method for parallelizing training of deep neural networks on large GPU clusters (https://arxiv.org/abs/1404.5997). In particular, when batch size is changed to k * batch_size, the learning rate is changed to learning_rate * sqrt(k). Prior works have used this square root scaling for fixed batch sizes (i.e., batch size is determined at the start of training based on the number of GPUs available) to find the optimal learning rate for a given batch size when the optimal learning rate is known for a different batch size. In this work, we find that this formulation works well for variable batch sizes also (i.e., changing learning rate for each batch based on the size of the batch). We thank the reviewer for pointing out this missing motivation, and we will add this to the final paper along with the relevant references.
>
> **Re: Since there are multiple components involved in the framework, laying out its scope in terms of tasks and models it would benefit would be really helpful for its usability in the real world.**
>
> BucketSampler is useful when fine-tuning is performed on datasets with variable length sequences. TokenDrop is useful on tasks where dropping stopwords from input sequences does not change the output labels. We will make this clear in the final paper.
>
> **Re: Typos Grammar Style And Presentation Improvements**
>
> We thank the reviewer for the suggestions. We will incorporate them in the final paper.

---

### Official Review · Reviewer_QVce · 2023-08-04

**Soundness:** 3

**Excitement:**

3: Ambivalent: It has merits (e.g., it reports state-of-the-art results, the idea is nice), but there are key weaknesses (e.g., it describes incremental work), and it can significantly benefit from another round of revision. However, I won't object to accepting it if my co-reviewers champion it.

**Paper Topic And Main Contributions:**

This paper presents two techniques to improve the efficiency of LLM fine-tuning. The first one is to randomly drop out a subset of stop words in each sequence in every epoch to reduce the number of tokens to encode/decode and alleviate overfitting. The second one is based on  bucketing, which include several improvements: 1) batch size is determined based on hardware limit and bucket sequence length, 2) residual buckets are merged together, 3) adaptive learning rate based on batch size. Empirical evaluations show the combination of the two techniques can significantly speedup the fine-tuning without loss of accuracy, comparing to a random sampler baseline.

**Questions For The Authors:**

1. Figure 1 is used to show that there is still overfitting during fine-tuning without Dropout. It is also claimed that TokenDrop is better in preventing overfitting. Are there any statistics/charts to support this?
2. Line 186, it states that “Unlike Dropout, TokenDrop introduces data diversity between the different training epochs”. This sounds inaccurate to me, since Dropout also causes data diversity by random sampling in different training epochs.
3. Algorithm 1, Line 8: is this line incomplete?
4. Algorithm 3, Line 4: does this mean for non-residual batches, TokenDrop is not applied? If that’s true, what’s the ratio of tokens dropped during fine-tuning?

**Reasons To Accept:**

1. This paper studies the improvement of an important task. Fine-tuning LLM is the most widely adapted way to leveraging LLMs. Improving the fine-tuning efficiently can have great impact in both academia and industry.
2. The proposed techniques can help both LLM fine-tuning and inference. Especially for the real-time inference scenario, the experiments show after the fine-tuning, dropping stop words during inference time can bring a speedup of 2.2X and almost without accuracy loss.

**Reasons To Reject:**

1. It would be good to show some more detailed analyses. For example, how does the speedup change with different fine-tuning set sizes, different length distributions?
2. The random sampler baseline used in empirical evaluation is weak. Maybe a simple sequence length bucketing baseline (e.g., `LengthGroupedSampler`, or `tf.bucket_by_sequence_length` without varying batch sizes) could be a better baseline?
3. The ablation study does not clearly reveal the contribution from TokenDrop.

**Reproducibility:**

3: Could reproduce the results with some difficulty. The settings of parameters are underspecified or subjectively determined; the training/evaluation data are not widely available.

**Reviewer Confidence:**

3: Pretty sure, but there's a chance I missed something. Although I have a good feel for this area in general, I did not carefully check the paper's details, e.g., the math, experimental design, or novelty.

---

> ### Author Rebuttal · Authors · 2023-08-29
>
> We thank the reviewer for the insightful review. We address the questions and comments below.
>
> **Re:  How does the speedup change with different fine-tuning set sizes, different length distributions?**
>
> **Dataset size**: Speedup is typically higher on larger datasets, since the overheads introduced by TokenDrop+BucketSampler are better amortized over the course of fine-tuning. For instance, the overheads of TokenDrop+BucketSampler account for approximately 20% of the wall-clock fine-tuning time when training on RTE, which has only 2.5K training samples. On the other hand, when fine-tuning on MNLI, which is a relatively large dataset with 393K training samples, the overheads of TokenDrop+BucketSampler account for only around 5% of the wall-clock fine-tuning time. This can be seen in Fig. 5 in Appendix C (page 11) in the paper. We also note that in spite of the overheads, TokenDrop+BucketSampler achieves substantial speedups over prior methods even on small datasets.
>
> **Length distribution**: Potential for speedup is maximum when a majority of sequences in the dataset are short, but there are a substantial number of long sequences as well. For example, in the SST-2 dataset, approximately 80% of all sequences have length <20, but approximately 10% of all sequences have length >35. As a result, when using RandomSampler with a batch size of 64, there is a >99% chance that each batch will have at least one sequence with length >35, leading to all samples in the batch being padded to >35 tokens. In contrast, when using TokenDrop+BucketSampler, the majority of batches can be processed with length <20, leading to large speedups over RandomSampler. This can be seen in Table 1, where speedup is maximum on SST-2. We will add a detailed analysis in the final paper about the correlation between the sequence length spread and size of a dataset, and the speedup achieved.
>
> **Re: Stronger baselines (e.g., LengthGroupedSampler, or tf.bucket_by_sequence_length without varying batch sizes).**
>
> We compare TokenDrop+BucketSampler with LengthGroupedSampler and tf.bucket_by_sequence_length with variable batch sizes in Table 4 (page 8 in our paper) in terms of both accuracy and efficiency. In summary, we find that TokenDrop+BucketSampler achieves ~0.3% higher accuracy along with 1.6X speedup over LengthGroupedSampler, and 5% higher accuracy along with 1.1X speedup over tf.bucket_by_sequence_length with variable batch sizes (batch_size = HWCap for each bucket) averaged across the GLUE tasks and SQUAD with Roberta-Base. Following the reviewer’s suggestion, we performed a new experiment and found that TokenDrop+BucketSampler achieves ~0.3% higher accuracy along with 1.5X speedup over tf.bucket_by_sequence_length without variable batch sizes averaged across the GLUE tasks and SQUAD. We can also report speedups over these methods for each task (similar to Table 1) in the final paper. We chose RandomSampler as baseline for the primary results since it is by far the most widely used method in practice for fine-tuning LLMs.
>
> **Re: The ablation study does not clearly reveal the contribution from TokenDrop.**
>
> Following the reviewer’s suggestion, we have evaluated TokenDrop’s contribution. When using only BucketSampler, the average accuracy is 81.67% and speedup (over RandomSampler) is 5.1X on GLUE and SQUAD with Roberta-Base. When both TokenDrop and BucketSampler are used, the accuracy and speedup increase to 81.98% and 5.9X, respectively. Therefore, the use of TokenDrop clearly improves both the accuracy and efficiency of fine-tuning. We thank the reviewer for pointing out this missing entry, and we will add this to Table 4 in the final paper.
>
> **Re: Figure 1 is used to show that there is still overfitting during fine-tuning without Dropout. It is also claimed that TokenDrop is better in preventing overfitting. Are there any statistics/charts to support this?**
>
> As we are unable to attach figures to the rebuttal, we use the table below to compare the training and validation losses with and without TokenDrop while fine-tuning Roberta-Base on RTE. The training loss decreases more gradually when TokenDrop is used, since different data samples are presented in each epoch. As a result, the model does not simply memorize the training data within a few epochs, leading to better generalization performance. We will add graphs similar to Figure 1 to demonstrate that TokenDrop does indeed reduce the impact of overfitting when fine-tuning on datasets of different sizes in the final paper using both Roberta-Base and Roberta-Large.
>
> |          | Without TokenDrop   |              | With TokenDrop |            |
> |--------| -------- | ------- | ------- | ------- |
> | Epoch| Train Loss | Validation Loss | Train Loss | Validation Loss |
> | 1 | 0.59 | 0.83| 0.57 | 0.79|
> | 2 | 0.32 | 0.68 | 0.42 | 0.57|
> |3| 0.16|0.59|0.28|0.5|
> |4|0.11|0.61|0.19|0.47|
> |5|0.09|0.69|0.17|0.51|
> |6|0.08|0.77|0.14|0.52|
> |7|0.06|0.98|0.1|0.54|
> |8|0.04|1.16|0.09|0.61|
> |9|0.04|1.36|0.08|0.74|
> |10|0.03|1.39|0.05|0.87|
>
> **Re: Line 186, it states that “Unlike Dropout, TokenDrop introduces data diversity between the different training epochs”. This sounds inaccurate to me, since Dropout also causes data diversity by random sampling in different training epochs.**
>
> Dropout only randomly drops a random subset of neurons in each training iteration. The training set is not changed between epochs. On the other hand, TokenDrop changes the training set itself in each epoch, by dropping a random subset of stopwords from each sample. This is what we meant by data diversity between different epochs, since a different subset of stopwords is dropped from each training sample in every epoch, leading to different training samples being presented to the model in every epoch. We will clarify this in the paper.
>
> **Re: Algorithm 1, Line 8: is this line incomplete?**
>
> Yes, it should be Tokens_to_drop(sample)--. We thank the reviewer for pointing out this mistake, and we will fix it in the paper.
>
> **Re: Algorithm 3, Line 4: does this mean for non-residual batches, TokenDrop is not applied? If that’s true, what’s the ratio of tokens dropped during fine-tuning?**
>
> TokenDrop is applied to non-residual batches also. Since batches are only formed from sequences within the same bucket, all sequences in a batch will have min_seq_len(bucket) < length < max_seq_len(bucket). Therefore, if a sequence in a non-residual batch has length greater than the minimum length of all sequences in that batch, then {length - min_seq_len(batch)} tokens are dropped from that sequence. In our experiments, we define buckets in length increments of 5, i.e, bucket 1 has (min_seq_len, max_seq_len) = [0,5), bucket 2 has (min_seq_len, max_seq_len) = [5,10) and so on. If, for example, a batch drawn from bucket 2 has 4 sequences with lengths (5, 6, 7 and 9), then the number of tokens dropped from these sequences are (0, 1, 2, 4), respectively. The ratio of tokens dropped varies across datasets, and is typically between 15-20% of all tokens. We will make this clearer in the final version of the paper.

---

### Meta-Review · Area_Chair_8ooK · 2023-09-10

**Recommendation:** 3

**Metareview:**

This paper presents two methods to improve the efficiency of fine-tuning, one relates to randomly dropping some of the input tokens, and the second is based on bucketing.  The reviewers appreciated the quality of the results, and the ablations. Some of the reviewers initially had some concerns around the analysis and the baselines, as well as the scope of the experiments. The rebuttal helped alleviate many of these concerns.

---

### Decision · Program_Chairs · 2023-10-07

**Decision:**

Accept-Findings

**Comment:**

This paper presents two methods to improve the efficiency of fine-tuning, one relates to randomly dropping some of the input tokens, and the second is based on bucketing.  The reviewers appreciated the quality of the results, and the ablations. Some of the reviewers initially had some concerns around the analysis and the baselines, as well as the scope of the experiments. The rebuttal helped alleviate many of these concerns.